# KV Prediction for Improved Time to First Token

## Abstract

Inference with transformer-based language models begins with a prompt processing step. In this step, the model generates the first output token and stores the KV cache needed for future generation steps. This prompt processing step can be computationally expensive, taking 10s of seconds or more for billion-parameter models on edge devices when prompt lengths or batch sizes rise. This degrades user experience by introducing significant latency into the model's outputs. To reduce the time spent producing the first output (known as the "time to first token", or *TTFT*) of a pretrained model, we introduce a novel method called KV Prediction. In our method, a small *auxiliary* model is used to process the prompt and produce an approximation of the KV cache used by a *base* model. This approximated KV cache is then used with the base model for autoregressive generation without the need to query the auxiliary model again. We demonstrate that our method produces a pareto-optimal efficiency-accuracy trade-off when compared to baselines. On TriviaQA, we demonstrate relative accuracy improvements in the range of $15\% - 50\%$ across a range of TTFT FLOPs budgets. We also demonstrate accuracy improvements of up to $30\%$ on HumanEval python code completion at fixed TTFT FLOPs budgets. Additionally, we benchmark models on an Apple M2 Pro CPU and demonstrate that our improvement in FLOPs translates to a TTFT speedup on hardware. We release our code for reproducibility.

## 1 Introduction

Large language models (LLMs) have demonstrated impressive capabilities on many downstream tasks (Gunter et al., 2024; Achiam et al., 2023; Chowdhery et al., 2022; Abdin et al., 2024). However, the high computational cost of running large language models results in limited capabilities for on-device inference. On-device inference is essential for privacy, latency, energy efficiency, and performance in limited-connectivity areas (Frantar et al., 2022; Alizadeh-Vahid et al., 2023; Stojkovic et al., 2024). For these reasons, LLM efficiency remains an important and active area of research.

LLM inference with the popular transformer (Vaswani et al., 2017) architecture consists of two phases. First, in the *prompt processing* phase, the model processes an input prompt to populate the KV cache. Next, in the *generation* phase, the model autoregressively generates output tokens. Many recent works focus on improving generation time through approximations of the KV cache (Wu & Tu, 2024; Jiang et al., 2023a; Ge et al., 2023a;b; Li et al., 2023), but only a few recent works have explored improving the prompt processing time (Fu et al., 2024).

Improving prompt processing time allows an application using the LLM to begin sending outputs to the user earlier. The "time to first token" (TTFT) refers to the length of time between a user's input query and the production of the first output token. In scenarios such as chatting with a user, the TTFT may be a more important runtime experience metric than autoregressive generation time, since the user can begin consuming outputs after the first token is produced. For on-device models, prompt processing times can be intolerably slow (up to 10s of seconds, Fig. 1). Reducing TTFT in these cases enables a better user experience.

We present a method to improve TTFT by processing the prompt with a small *auxiliary* transformer model. Our method runs the auxiliary model and stores its KV cache. It then uses a learned linear projection to predict the KV cache of another transformer model (the *base* model) using only the

KV cache of the auxiliary model as input. After the KV cache is predicted, inference continues exclusively with the base model. As a result, no runtime overhead is introduced during generation.

We demonstrate that our method improves over the efficiency-accuracy trade-off achievable by our baselines. For example, on TriviaQA (Joshi et al., 2017), our method improves accuracy retention by $15\% - 50\%$ compared to baselines at equal TTFT FLOP counts. On HumanEval (Chen et al., 2021) python code completion, we demonstrate accuracy improvements up to $30\%$ over baselines at equal TTFT FLOP counts. We also provide on-device timing experiments to demonstrate that our FLOPs gains translate to on-device runtime improvements.

Our contributions are as follows: (1) We develop a novel method called KV Prediction for using a smaller auxiliary model to predict the KV cache of a larger base model. (2) We show that our model produces a stronger efficiency-accuracy trade-off compared to baselines. (3) We analyze the runtime characteristics of KV Prediction models on-device. Additionally, we release our code for reproducibility.

The rest of the paper is organized as follows. In Section 2, we give an overview of related work. In Section 3, we motivate our work, demonstrating that TTFT can be problematically high during on-device inference. In Section 4, we describe our KV Prediction method. In Section 5, we describe our experimental setup for evaluating KV Prediction. In Section 6, we give our main results, showing that our method obtains the pareto-optimal efficiency-accuracy trade-off. In Section 7, we analyze our predicted KV caches. In Section 8, we conclude.

## 2 RELATED WORK

**On-Device TTFT Efficiency:** Few works have explored our problem domain of improving on-device TTFT. LazyLLM (Fu et al., 2024) uses an attention-based token dropping strategy to drop unneeded tokens at inference time. These tokens can be revived later during the generation phase. Random token dropping (Yao et al., 2022) and static token pruning are both studied in Fu et al. (2024) as methods for improving TTFT. These methods are our baselines.

**Server-Side TTFT Efficiency:** Cachegen (Liu et al., 2023) compresses KV caches for faster TTFT, but their setup assumes that a precomputed, pre-compressed KV cache is stored on a server that is available over network. Another similar server-based approach, CritiPrefill (Lv et al., 2024), performs attention-based pruning of KV cache segments on a per-query basis. Sarathi (Agrawal et al., 2023) reduces pipeline bubbles in batched server-side inference. Etalon (Agrawal et al., 2024) provides a framework for evaluating server-side inference performance, and analyses various methods. Our investigation differs from these in that we focus on on-device TTFT improvements. Other works focus on efficient long-context processing at extreme context lengths (Gao et al., 2024; Jiang et al., 2024; Xiong et al., 2023), which is currently not feasible on-device.

**Context Compression:** Previous works have investigated compressing the KV cache for improving generation efficiency. PyramidKV (Cai et al., 2024), StreamingLLM (Xiao et al., 2023), SnapKV (Li et al., 2024), and Model Tells You What To Discard (Ge et al., 2023a) all compress the KV cache along the token dimension by observing attention scores and pruning irrelevant tokens. However, their methods compute a forward pass before pruning. Thus, they improve generation time, but not TTFT. Layer-Condensed KV Cache (Wu & Tu, 2024) compresses an $N$-layer KV cache into a 1-layer KV cache, but requires 9 prompt processing steps and negatively impacts TTFT. Other works such as KIVI (Liu et al., 2024) and GEAR (Kang et al., 2024) have explored quantizing the KV cache, but do not improve TTFT.

A few recent works have explored context compression with a separate network to improve generation time (Jiang et al., 2023a;b; Ge et al., 2023b; Li et al., 2023). As shown in Fu et al. (2024), such methods can increase TTFT due to more expensive prompt processing.

**Sharing Activations Between Models:** Similar to our method, Tandem Transformers (AishwaryaP et al., 2024) uses two different models that share hidden activations. However, their focus is on improving the performance of Speculative Decoding (Leviathan et al., 2022), not TTFT. Other methods explore using a larger teacher model to distill knowledge into a smaller student model to improve efficiency (Bagherinezhad et al., 2018; Gou et al., 2020; Xu et al., 2024). After training, the teacher model is discarded. Our method differs in that both models are retained.

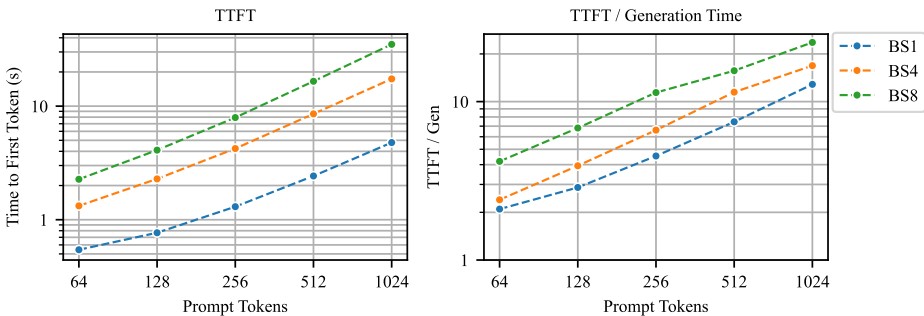

Figure 1: Time to First Token (TTFT) and ratio of TTFT to generation time for an OpenELM 3B model on an M2 Pro CPU (EveryMac, 2024) with 32GB of RAM. We evaluate at batch sizes 1, 4, and 8.

**General Techniques for Efficiency:** Many works have explored quantization (Lin et al., 2023; Tseng et al., 2024; Frantar et al., 2022; Dettmers et al., 2023; Shao et al., 2023; Egiazarian et al., 2024), pruning (Alizadeh-Vahid et al., 2023; Ma et al., 2023; Zheng et al., 2024), and efficient design (Mehta et al., 2024; Zhang et al., 2024; Abdin et al., 2024) to improve LLM efficiency. These techniques are orthogonal to ours, as they can be combined with our method or our baselines.

## 3 MOTIVATION: TIME TO FIRST TOKEN (TTFT)

We observe that the TTFT of running a language model on an edge device can be prohibitively high, negatively impacting user experience. We visualize total prompt processing time of OpenELM (Mehta et al., 2024) models on an M2 Pro CPU (EveryMac, 2024) with 32GB of RAM in Fig. 1. As prompt lengths and batch sizes increase, the TTFT transitions from noticeable to intolerable. Most users are not willing to wait more than a few seconds for a response from a chat bot (Dashly, 2023), but on-device prompt processing times can far exceed this. This long processing time would be further exacerbated by running the model on low-end hardware instead of a high-end laptop CPU.

In addition to the negative user experience created by high TTFTs, we note that the ratio of prompt processing time to generation time is of particular significance to applications that generate short responses. For example, in a question-answering system, the model may only generate a few tokens. Thus, the fractional runtime spent during prompt processing can dominate the overall runtime. In Fig. 1, we visualize the ratio of TTFT to the autoregressive generation time for an OpenELM 3B model. As prompt length and batch size increases, the process becomes compute-bound and the cost of prompt processing substantially increases relative to the cost of a generation step.

To illustrate the impact of this high ratio of prompt processing time to generation time, consider a question-answering system with 1024 tokens of context, operating at batch size 1, and generating a 2 token response. In this case, the model will spend 4.8 seconds processing the prompt and generating the first token, and only $4.8/12 = 0.4$ seconds generating the second token (since prompt processing takes 12 times as long as generation, see Fig. 1). In this case, 5.2 seconds are spent generating the output, of which $4.8/5.2 = 92.3\%$ is spent in prompt processing.

In summary, large prompt processing times can negatively impact practical usage of a model in 2 ways. First, they can result in a large TTFT, negatively impacting user experience. Second, they also can represent a large fraction of the total compute used in inference. Our goal is to reduce on-device TTFT through a novel modeling strategy which we describe in the next section.

## 4 KV PREDICTION

Recent works have shown that the KV cache can be compressed with little or no loss in accuracy at generation time (Cai et al., 2024; Li et al., 2024; Xiao et al., 2023; Ge et al., 2023a). Thus, we hypothesize that the KV cache for a model can be approximated efficiently to reduce on-device TTFT. Our approximation uses an efficient learned model to predict the KV cache.

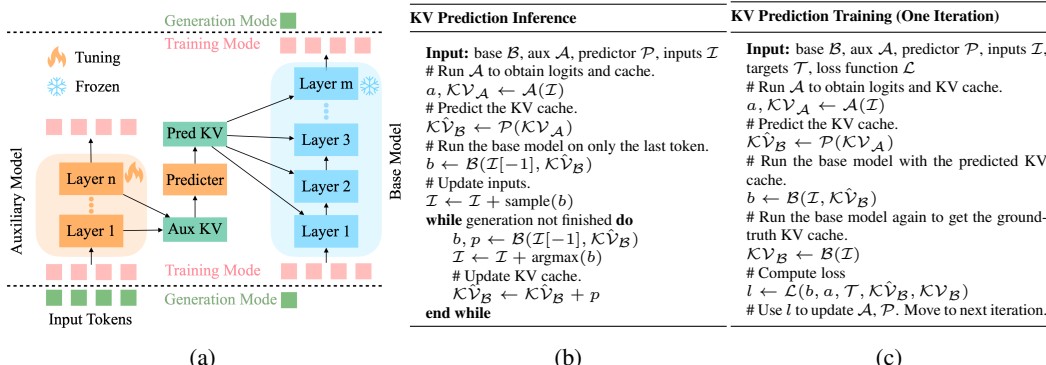

Figure 2: (a) An overview of our method. (b) Our inference method. (c) Our training method.

Our intuition is that an accurate KV cache prediction method should use a model of similar structure to the base model, so that the structure of the KV cache can be modeled more accurately. Thus, we use a smaller learned auxiliary transformer to process the prompt. Then, a set of learned linear projections is used to predict the base model's KV cache using the auxiliary model's KV cache as input. Fig. 2a shows an overview of our method.

In the following subsections, we present details of our method. In Section 4.1, we give an overview of training and inference. In Section 4.2, we provide more architectural details of our KV Prediction models. In Section 4.3, we describe our loss function. Finally, in Section 4.4, we analyze the reduction in TTFT.

## 4.1 Predicting KV Caches

Our model contains a frozen pretrained base network $\mathcal{B}$, a learned auxiliary network $\mathcal{A}$, and a learned KV predictor $\mathcal{P}$. The auxiliary and predictor networks are used to generate the predicted KV cache efficiently. Afterwards, inference proceeds with the base model.

**Inference:** During inference (Fig. 2b) the prompt is passed to the auxiliary model and the auxiliary KV cache $\mathcal{KV}_\mathcal{A}$ is computed. Then, the predicted base KV cache $\hat{\mathcal{KV}}_\mathcal{B} = \mathcal{P}(\mathcal{KV}_\mathcal{A})$ is computed. At this point, $\mathcal{A}$ and $\mathcal{P}$ are no longer required to continue inference. To generate the first token, a single-token generation step of $\mathcal{B}$ is run, but with $\hat{\mathcal{KV}}_\mathcal{B}$ being used as the keys and values in the attention operations (instead of using the keys and values from the base model's QKV projections). The logits produced from this generation step are used to produce the first output token. Now that the first token has been produced, generation continues autoregressively in the standard fashion, with new KV cache entries being added as new tokens are processed.

**Training:** During training (Fig. 2c), a sequence of tokens $\mathcal{I}$ are fed to the auxiliary model $\mathcal{A}$ to produce output logits $\mathcal{A}(\mathcal{I})$ and a KV cache $\mathcal{KV}_\mathcal{A}$. Then, the predicted KV cache $\hat{\mathcal{KV}}_\mathcal{B} = \mathcal{P}(\mathcal{KV}_\mathcal{A})$ is computed. Finally, a forward pass of $\mathcal{B}$ is computed using the predicted KV cache $\hat{\mathcal{KV}}_\mathcal{B}$ (instead of using the keys and values from the base model's QKV projections) to produce output logits $\mathcal{B}(\mathcal{I})$. In other words, the base model computes a forward pass using masked cross-attention with the predicted KV cache to produce output logits. Errors in the base model's logits backpropagate through the frozen base model and into $\mathcal{A}$ and $\mathcal{P}$.

## 4.2 Architectural Details

Our KV Prediction models are fully specified by our base, auxiliary, and predictor networks. Our base network always consists of a standard pretrained frozen transformer network. Here we describe the architectural details of our auxiliary and predictor networks.

**Auxiliary Networks:** We use two different methods for choosing an auxiliary network. In the first method, referred to as KVP-C (KV Prediction with a Canonical model), we choose the auxiliary network to be a smaller model in the same family as the base model. In the second method for choosing

| Model | Aux | $j_0$ | $j_1$ | $j_2$ | $j_3$ | $j_4$ | $j_5$ | $j_6$ | $j_7$ | $j_8$ | $j_9$ | $j_{10}$ | $j_{11}$ | $j_{12}$ | $j_{13}$ | $j_{14}$ | $j_{15}$ | $j_{16}$ | $j_{17}$ | $j_{18}$ | $j_{19}$ | $j_{20}$ | $j_{21}$ | $j_{22}$ | $j_{23}$ | $j_{24}$ | $j_{25}$ | $j_{26}$ | $j_{27}$ |
|---|---|---|---|---|---|---|---|---|---|---|---|---|---|---|---|---|---|---|---|---|---|---|---|---|---|---|---|---|---|
| OE1.1B-KVP-C-450M | OE450M | 0 | 0 | 1 | 1 | 2 | 2 | 3 | 3 | 4 | 4 | 5 | 5 | 6 | 6 | 7 | 7 | 8 | 9 | 10 | 11 | 12 | 13 | 14 | 15 | 16 | 17 | 18 | 19 |
| OE1.1B-KVP-C-270M | OE270M | 0 | 0 | 1 | 1 | 2 | 2 | 3 | 3 | 4 | 4 | 5 | 5 | 6 | 6 | 7 | 7 | 8 | 8 | 9 | 9 | 10 | 10 | 11 | 11 | 12 | 13 | 14 | 15 |
| OE1.1B-KVP-LP-0.75 | OE-LP-0.75 | 0 | 1 | 2 | 2 | 3 | 4 | 5 | 5 | 6 | 7 | 8 | 8 | 9 | 10 | 11 | 11 | 12 | 13 | 14 | 14 | 15 | 16 | 17 | 17 | 18 | 19 | 20 | 20 |
| OE1.1B-KVP-LP-0.50 | OE-LP-0.50 | 0 | 0 | 1 | 1 | 2 | 2 | 3 | 3 | 4 | 4 | 5 | 5 | 6 | 6 | 7 | 7 | 8 | 8 | 9 | 9 | 10 | 10 | 11 | 11 | 12 | 12 | 13 | 13 |
| OE1.1B-KVP-LP-0.25 | OE-LP-0.25 | 0 | 0 | 0 | 0 | 1 | 1 | 1 | 1 | 2 | 2 | 2 | 2 | 3 | 3 | 3 | 3 | 4 | 4 | 4 | 4 | 5 | 5 | 5 | 5 | 6 | 6 | 6 | 6 |

(a) KV Prediction models using a base model of OpenELM 1.1B.

| Model | Aux | $j_0$ | $j_1$ | $j_2$ | $j_3$ | $j_4$ | $j_5$ | $j_6$ | $j_7$ | $j_8$ | $j_9$ | $j_{10}$ | $j_{11}$ | $j_{12}$ | $j_{13}$ | $j_{14}$ | $j_{15}$ | $j_{16}$ | $j_{17}$ | $j_{18}$ | $j_{19}$ | $j_{20}$ | $j_{21}$ | $j_{22}$ | $j_{23}$ | $j_{24}$ | $j_{25}$ | $j_{26}$ | $j_{27}$ | $j_{28}$ | $j_{29}$ | $j_{30}$ | $j_{31}$ | $j_{32}$ | $j_{33}$ | $j_{34}$ | $j_{35}$ |
|---|---|---|---|---|---|---|---|---|---|---|---|---|---|---|---|---|---|---|---|---|---|---|---|---|---|---|---|---|---|---|---|---|---|---|---|---|---|
| OE3B-KVP-C-1.1B | OE1.1B | 0 | 0 | 1 | 1 | 2 | 2 | 3 | 3 | 4 | 4 | 5 | 5 | 6 | 6 | 7 | 7 | 8 | 9 | 10 | 11 | 12 | 13 | 14 | 15 | 16 | 17 | 18 | 19 | 20 | 21 | 22 | 23 | 24 | 25 | 26 | 27 |
| OE3B-KVP-C-450M | OE450M | 0 | 0 | 1 | 1 | 2 | 2 | 3 | 3 | 4 | 4 | 5 | 5 | 6 | 6 | 7 | 7 | 8 | 8 | 9 | 9 | 10 | 10 | 11 | 11 | 12 | 12 | 13 | 13 | 14 | 14 | 15 | 15 | 16 | 17 | 18 | 19 |
| OE3B-KVP-C-270M | OE270M | 0 | 0 | 0 | 1 | 1 | 1 | 2 | 2 | 2 | 3 | 3 | 3 | 4 | 4 | 4 | 5 | 5 | 5 | 6 | 6 | 6 | 7 | 7 | 7 | 8 | 8 | 8 | 9 | 9 | 9 | 10 | 10 | 11 | 12 | 14 | 15 |
| OE3B-KVP-LP-0.75 | OE-LP-0.75 | 0 | 1 | 2 | 2 | 3 | 4 | 5 | 5 | 6 | 7 | 8 | 8 | 9 | 10 | 11 | 11 | 12 | 13 | 14 | 15 | 16 | 17 | 17 | 18 | 19 | 20 | 20 | 21 | 22 | 23 | 24 | 25 | 25 | 24 | 25 | 26 |
| OE3B-KVP-LP-0.50 | OE-LP-0.50 | 0 | 0 | 1 | 1 | 2 | 2 | 3 | 3 | 4 | 4 | 5 | 5 | 6 | 6 | 7 | 7 | 8 | 8 | 9 | 9 | 10 | 10 | 11 | 11 | 12 | 12 | 13 | 13 | 14 | 14 | 15 | 15 | 16 | 16 | 17 | 17 |
| OE3B-KVP-LP-0.25 | OE-LP-0.25 | 0 | 0 | 0 | 0 | 1 | 1 | 1 | 1 | 2 | 2 | 2 | 2 | 3 | 3 | 3 | 3 | 4 | 4 | 4 | 4 | 5 | 5 | 5 | 5 | 6 | 6 | 6 | 6 | 7 | 7 | 7 | 7 | 8 | 8 | 8 | 8 |

(b) KV Prediction models using a base model of OpenELM 3B.

Table 1: KV Prediction models. Each model is specified by a base network, an auxiliary network, and the mapping of input auxiliary layers $j_i$ to base layers $i$. (Table 1a): models using a base network of OpenELM 1.1B. (Table 1b): models using a base network of OpenELM 3B.

| $\lambda_C$ | 0 | 1/28 | 1 |
|---|---|---|---|
| Acc | 8.95 | **17.38** | 15.96 |

Table 2: Ablation on TriviaQA (1-shot) of our choice of consistency loss coefficient $\lambda_C$. Our model is OpenELM1.1B-KVP-LP-0.50.[1]

an `auxiliary` network, referred to as `KVP-LP` (KV Prediction with a Layer-Pruned model), our auxiliary network consists of a copy of the base network with some of the layers removed.

**Predictor Networks:** For our *predictor* network, we use a simple set of learned linear transforms to predict each layer of the base model's KV cache independently. We choose linear transforms primarily for their efficiency, as our focus is on reducing TTFT.

First, we need to choose which layer of the auxiliary cache to use to predict each layer of the base cache. To do this, we define a mapping from auxiliary cache layers $j$ to base cache layers $i$. Let $n_\mathcal{B}$ be the number of transformer layers in the base network, and $n_\mathcal{A}$ be the number of transformer layers in the auxiliary network. For each $i \in [0, ..., n_\mathcal{B} - 1]$, let $j_i \in [0, ..., n_\mathcal{A} - 1]$ denote the $j_i$th layer of the auxiliary cache $\mathcal{KV}_{\mathcal{A}j_i}$. We will use $\mathcal{KV}_{\mathcal{A}j_i}$ as input to a linear function to predict the layer of the base cache $\hat{\mathcal{KV}}_{\mathcal{B}i}$.

Now that we have defined our mapping $j_i$, we define our set of linear functions. Our linear functions are defined as $\mathcal{F}_i(\mathcal{KV}_{\mathcal{A}j_i}) : \mathcal{R}^{d_{\mathcal{KV}_{\mathcal{A}j_i}}} \to \mathcal{R}^{d_{\mathcal{KV}_{\mathcal{B}i}}}$, where $d_{\mathcal{KV}_{\mathcal{A}j_i}}, d_{\mathcal{KV}_{\mathcal{B}i}}$ are the feature dimensions of the auxiliary and base KV caches at layers $j_i$ and $i$, respectively. We use these linear functions to compute each layer of the base KV cache, $\hat{\mathcal{KV}}_{\mathcal{B}i} = \mathcal{F}_i(\mathcal{KV}_{\mathcal{A}j_i})$. Once each layer $\hat{\mathcal{KV}}_{\mathcal{B}i}$ is predicted, we concatenate them to produce $\hat{\mathcal{KV}}_{\mathcal{B}}$.

When choosing the auxiliary KV cache layer $j_i$ to use as the input to $\mathcal{F}_i$, we build on top of the observation in Brandon et al. (2024) that neighboring transformer layers will have KV caches that are more similar than distant layers. Thus, we use the first layer of $\mathcal{KV}_\mathcal{A}$ as input to predict the first several layers of $\mathcal{KV}_\mathcal{B}$, and the second layer of $\mathcal{KV}_\mathcal{A}$ as input to predict the next several layers of $\mathcal{KV}_\mathcal{B}$, and so forth. When $n_\mathcal{A}$ does not divide evenly into $n_\mathcal{B}$, we perform rounding.

**OpenELM KV Prediction Specification:** When experimenting with KV Prediction, we use OpenELM (Mehta et al., 2024) models. We choose OpenELM because of its variable KV cache size in each layer, allowing us to evaluate our method in this challenging scenario.

For KVP-C experiments, we use smaller models from the OpenELM family as the auxiliary models. For KVP-LP experiments, we define a set of layer-pruned OpenELM models in Appendix A, which we use as the auxiliary model.

Our KV Prediction models are listed in Table 1. Each model is specified by a base network, an auxiliary network, and the pattern of input auxiliary layers $j_i$ used to predict base KV cache layer $i$. When referring to a KV prediction architecture, we specify its base network, followed by either KVP-C or KVP-LP, followed by its auxiliary network (e.g. OE1.1B-KVP-C-450M).

### 4.3 Loss Function

In this subsection we describe our training loss. It consists of three components: the base loss $\mathcal{L}_B$, the auxiliary loss $\mathcal{L}_A$, and the consistency loss $\mathcal{L}_C$.

**Base loss:** The base loss $\mathcal{L}_B = \mathcal{C}_B(\mathcal{B}(\mathcal{I}), \mathcal{T})$ is the cross-entropy loss between the base model's outputs and the ground-truth labels $\mathcal{T}$. The base model is frozen, but the gradients flow backwards through the KV predictor and the auxiliary model. This loss helps ensure that the predicted KV cache is compatible with the base model.

**Auxiliary loss:** The auxiliary loss $\mathcal{L}_A = \mathcal{C}_A(\mathcal{A}(\mathcal{I}), \mathcal{T})$ is the cross-entropy loss between the auxiliary model's outputs and the ground-truth labels. Since the auxiliary logits $\mathcal{A}(\mathcal{I})$ are never used during inference, this loss is not strictly required to produce an auxiliary model and a predictor model that can support KV prediction. However, in preliminary experiments we find that $\mathcal{L}_A$ improves training convergence slightly.

**Consistency loss:** The consistency loss $\mathcal{L}_C = \mathbb{L}_1(\hat{\mathcal{KV}}_\mathcal{B}, \mathcal{KV}_\mathcal{B})$ is used to align the predicted KV caches with the base model's KV caches. The consistency loss is necessary to ensure that the predicted KV cache is compatible with the base model. To illustrate this point, we present an ablation in Table 2 showing a model trained with the consistency loss coefficient set to 0 ($\lambda_C = 0$). Without the consistency loss, the predicted KV cache performs poorly.

Our final loss is a simple weighted sum of the individual losses:

$$\mathcal{L}(\mathcal{B}(\mathcal{I}), \mathcal{A}(\mathcal{I}), \mathcal{T}, \hat{\mathcal{KV}}_\mathcal{B}, \mathcal{KV}_\mathcal{B}) = \lambda_B \mathcal{L}_B + \lambda_A \mathcal{L}_A + \lambda_C \mathcal{L}_C \tag{1}$$

$$= \lambda_B \mathcal{C}_A(\mathcal{B}(\mathcal{I}), \mathcal{T}) + \lambda_A \mathcal{C}_A(\mathcal{A}(\mathcal{I}), \mathcal{T}) + \lambda_C \mathbb{L}_1(\hat{\mathcal{KV}}_\mathcal{B}, \mathcal{KV}_\mathcal{B}) \tag{2}$$

We found that the loss terms $\mathcal{L}_B$ and $\mathcal{L}_A$ were balanced by simply setting $\lambda_B = \lambda_A = 1$. To choose $\lambda_C$, we perform an ablation between summing the loss across layers ($\lambda_C = 1$) and averaging the loss across layers ($\lambda_C = 1/n_B$) in Table 2. We found that $\lambda_C = 1/n_B$ performed better.

### 4.4 Runtime Analysis

**Improvements in TTFT FLOPs:** We analyze the improvement in TTFT of our method. The FLOPs-per-token compute cost of transformers inference can be estimated as $2P$, where $P$ is the number of parameters in the model (Kaplan et al., 2020). The total FLOPs required for prompt processing for $N$ tokens is $NP$. Thus, the ratio of the FLOPs of prompt processing to the ratio of FLOPs for a single generation step is $N$.

Let $t_\mathcal{N}(N)$ denote the forward pass FLOPs of network $\mathcal{N}$ with $N$ input tokens. As described in Section 4, producing the first output token using KV Prediction requires an N-token forward pass of $\mathcal{A}$, followed by an N-token forward pass of $\mathcal{P}$, then a single-token forward pass of $\mathcal{B}$ (to generate the first output token using the predicted KV cache). The FLOPs taken to process the prompt and generate the first token is $t_\mathcal{A}(N) + t_\mathcal{P}(N) + t_\mathcal{B}(1) = N t_\mathcal{A}(1) + t_\mathcal{P}(N) + t_\mathcal{B}(1)$. The computational cost of $t_\mathcal{P}(N)$ is negligible compared to $N t_\mathcal{A}(1)$, as it contains only a linear layer of smaller dimensionality than the transformer's FFN layers. For sufficient $N$, $N t_\mathcal{A}(1) \gg t_\mathcal{B}(1)$, and the FLOPs of a single inference are dominated by $N t_\mathcal{A}(1)$. Thus, the relative improvement in prompt processing FLOPs over the standard inference setup can be approximated as $t_\mathcal{A}(1)/t_\mathcal{B}(1)$.

**Memory Usage:** Our method requires deploying an auxiliary and predicter network in addition to the base network. The predicter is much smaller than a single transformer block, thus occupies negligible memory. The auxiliary network is generally a fraction of the size of the base network. Additionally, the auxiliary network can be unloaded from memory after prompt processing, as it is not needed during generation. To avoid cold starts on the next query, the auxiliary model can be reloaded into memory after inference. We do not employ this optimization in timing experiments.

---

[1]Results differ slightly from other tables, as training hyperparameters were not finalized during this ablation.

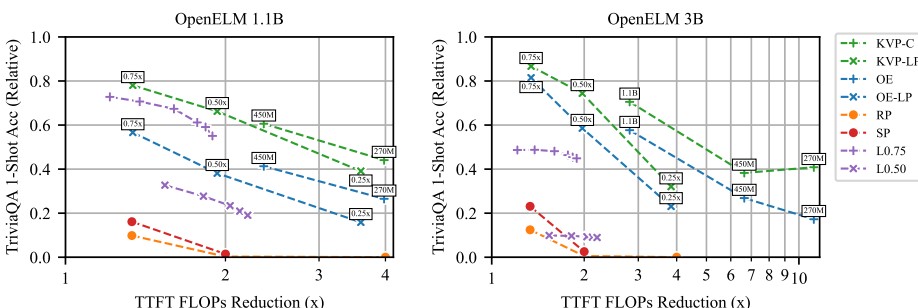

Figure 3: Efficiency-accuracy trade-off of our KV Prediction method (KVP-C, KVP-LP) on Trivi-aQA compared to baselines. The x-axis shows the relative reduction in FLOPs compared to the base network, and the y-axis shows the relative accuracy retention compared to the base network. (Left): Results using a base network of OpenELM 1.1B for KV Prediction. For KV Prediction models (green), points are annotated with the auxiliary network used. For example, the leftmost green "x" corresponds to OE1.1B-KVP-LP-0.75, and the leftmost green "+" corresponds to OE1.1B-KVP-C-450M. For OpenELM baselines (blue), points are annotated with the OpenELM variant used. All other baselines use variations of token pruning with different rates on OpenELM 1.1B (Right): Results using a base network of OpenELM 3B.

## 5 EXPERIMENTAL SETUP

We experiment with our KV Prediction method to evaluate the improvement in TTFT and the accuracy retention. Here we present details of our experimental setup.

**KV Prediction Models:** We experiment with KV Prediction using OpenELM (Mehta et al., 2024), as (1) it provides a suitable efficient architecture for on-device execution, and (2) its layers have variable KV cache sizes, allowing us to test our method's robustness to this challenging scenario.

To train our models, we reload the base and auxiliary model weights from pretrained OpenELM models (in the case of KVP-LP experiments, we only reload the unpruned layers into the auxiliary model). We follow the training hyperparameters of OpenELM, training on RefinedWeb (Penedo et al., 2023), ArXiv (Clement et al., 2019), and Wikipedia (Foundation, 2024). We shorten the training regime to 70k iterations on 64 H100 GPUs at a token length of 2048 and a batch size of 16, since we are reloading pretrained weights. For code completion experiments, our training setup is the same, but we instead train on The Stack (Kocetkov et al., 2022) and divide the learning rate by 10. A complete list of configs appears in our code release for reproducibility.

**Baselines:** Few works have explored methods for TTFT improvement applicable to on-device inference. Thus, our baselines follow Fu et al. (2024). Our first set of baselines uses token pruning. Our **RP** baseline consists of randomly pruning tokens from the input, sweeping across token retention ratios of [0.25, 0.50, 0.75]. Our **SP** baseline consists of probing the first 25% of network layers to obtain attention weights, then pruning tokens that have low attention scores and rerunning on the pruned tokens. We use token retention rates of [0.25, 0.50], omitting the higher retention rate of 0.75 since the overhead of processing 25% of the query with the unpruned input means that lower retention ratios are needed to obtain a similar speedup to random pruning. Our **L** baseline consists of LazyLLM (Fu et al., 2024). LazyLLM prunes at progressively higher rates through the network, from an initial higher retention rate (or low pruning rate) to a final lower retention rate (or higher pruning rate). We adopt the standard configuration suggested by the authors of beginning pruning 30% of the way into the network and ending pruning 90% of the way into the network. For **L0.75**, our beginning retention rate (which is active 30% of the way into the network) is 75%, and we sweep across ending retention rates of [0.75, 0.50, 0.25, 0.10, 0.05, 0.01]. For **L0.50**, our beginning retention rate is 0.50, and we sweep across end retention rates [0.50, 0.25, 0.10, 0.05, 0.01].

Our second set of baselines sweeps across model sizes in the OpenELM family. Our **OE** baseline consists of OpenELM 1.1B, OpenELM450M, and OpenELM270M. Our **OE-LP** baseline consists of OpenELM layer-pruned models, fine-tuned with the same settings as our KV Prediction models. See Appendix A for a detailed description of these layer-pruned architectures.

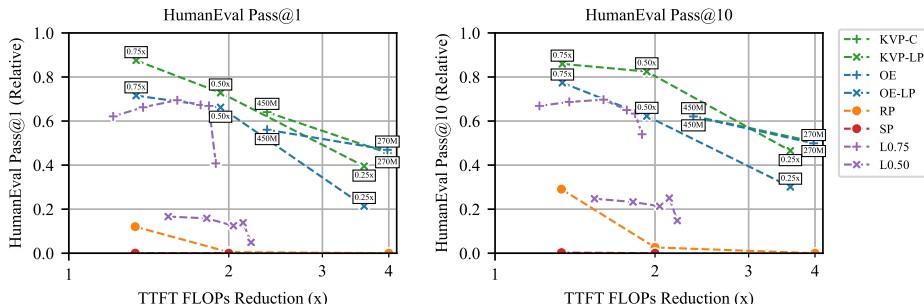

Figure 4: Efficiency-accuracy trade-off of our KV Prediction method (KVP-C, KVP-LP) compared to baselines on HumanEval python code completion. The x-axis shows the relative speedup in FLOPs compared to OpenELM 1.1B, and the y-axis shows the relative accuracy retention compared to OpenELM 1.1B. (Left): HumanEval Pass@1. (Right): HumanEval Pass@10 (Note that results for KVP-C and OE are overlapping.)

## 6 RESULTS

We analyze our method's performance on language modeling benchmarks in the following sections. Our main results are generative evaluations, which allow us to assess the compatibility of the initial predicted KV cache with the autoregressively generated KV cache. In Section 6.1, we investigate the improvement in TTFT of our method compared to baselines on question-answering with TriviaQA (Joshi et al., 2017), demonstrating that our method produces the pareto-optimal efficiency-accuracy tradeoff. In section Section 6.2, we demonstrate that our model also achieves the pareto-optimal efficiency-accuracy tradeoff on python code completion with HumanEval (Chen et al., 2021), supporting the generality of our method. Finally, in Section 6.3, we analyze the on-device runtime improvement in TTFT, demonstrating that our theoretical FLOPs reduction translates to runtime improvement on real hardware.

### 6.1 QUESTION-ANSWERING ON TRIVIAQA

We investigate our method's efficiency-accuracy trade-off for OpenELM 1.1B and OpenELM 3B in Fig. 3 (we also present these results in table format in Appendix B). We measure accuracy on TriviaQA using LLMEvalHarness (Gao et al., 2021), and use the FLOPs speedup approximation developed in Section 4.4.

Our method produces the strongest efficiency-accuracy trade-off, tracing a pareto-optimal curve. At a fixed FLOPs reduction, our method achieves the highest accuracy. Equivalently, at fixed accuracy, our method obtains the highest FLOPs reduction. The KVP-C strategy outperforms the KVP-LP method, but the KVP-LP method is able to achieve higher accuracy retention because larger models can be obtained. For instance, OE3B-KVP-LP-0.75 (Fig. 3, right side, top left point in the KVP-LP curve) has roughly $2\times$ as many parameters as OE3B-KVP-C-1.1B (Fig. 3, right side, top left point in the KVP-C curve). In all cases except OE3B-KVP-LP-0.25, our model produces higher accuracy at equivalent TTFT FLOPs compared to baselines.

Our OE baselines correspond to only using the auxiliary model architecture from the KVP-C experiment (but with different weights). Directly comparing our KVP-C method to these baselines, we see that our method strongly increases accuracy. A similar observation can be made when comparing OE-LP and KVP-LP models.

Our naive token-pruning baselines of random pruning (RP) and static pruning (SP) do not perform very well. As explored in Fu et al. (2024), SP can serve as a strong baseline when contexts are extremely long ($\sim 4k$) tokens. We conjecture that performance is worse here because there are fewer redundant tokens in TriviaQA evaluations. LazyLLM provides much higher accuracy retention than naive token-pruning, but accuracy decreases as we push to more aggressive FLOP reductions.

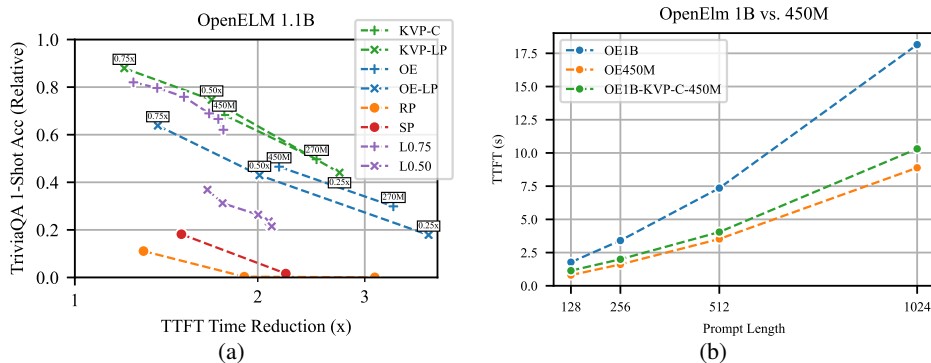

Figure 5: (Fig. 5a): Accuracy on the TriviaQA dataset compared to benchmarked time to first token on CPU. (Fig. 5b): The time to first token of our KVP prediction model OE1.1B-KVP-C-450M compared to OpenELM 1.1B and OpenELM450M.

## 6.2 CODE COMPLETION RESULTS

We investigate our method's efficiency-accuracy trade-off for code completion with a base model of OpenELM 1.1B in Fig. 4 (we also present these results in table format in Appendix B). We train our models on the Stack (Kocetkov et al., 2022) and measure performance on HumanEval's python code completion benchmark (Chen et al., 2021).

We find that our model produces the strongest efficiency-accuracy trade-off, tracing a pareto-optimal curve. As in the case of TriviaQA, our KVP-C strategy outperforms the KVP-LP method, but the KVP-LP method is able to achieve a higher accuracy retention because larger models can be chosen.

Our method obtains a much stronger efficiency-accuracy trade-off than OE and OE-LP baselines. Our KV prediction method also improves over token-pruning baselines (RP, SP, L0.75, and L0.50) in terms of efficiency and accuracy. The poor performance of token pruning methods can be attributed to the fact that code tokens have a tendency to carry less redundant information than general question-answering, making pruning more difficult.

## 6.3 TIMING EXPERIMENTS

We perform timing experiments to analyze the runtime improvement of our method. We measure the reduction in TTFT on an M2 Pro CPU (EveryMac, 2024) with 32GB of RAM using the average query length of TriviaQA 1-shot (59 tokens) and a batch size of 64. We plot this TTFT reduction (relative to the Base model) and the accuracy retention (relative to the base model) in Fig. 5a. We find that our method traces the pareto-optimal frontier. The TTFT of OE baseline models exceeds that of KVP-C models, but the stronger accuracy of the KVP-C models keeps them on the pareto-optimal frontier. A similar observation can be made for OE-LP and KVP-LP models. We also present these comparisons in table format in Appendix B.

Next, we measure the TTFT of a KV Prediction model compared to its base and auxiliary models. To do this, we perform a comparison of the runtime characteristics of OpenELM 1.1B, OE1.1B-KVP-C-450M, and OpenELM 450M in Fig. 5b. We set the batch size to 8 and show the change in TTFT as a function of prompt length. We find that the TTFT of OE1.1B-KVP-C-450M is within $10 - 20\%$ of the OpenELM 450M baseline, demonstrating that KV prediction has small overhead on-device relative to only running the auxiliary model. Both OE1.1B-KVP-C-450M and OE450M have a TTFT far superior to OpenELM 1.1B This indicates that our method is competitive with the auxiliary-only baseline in terms of speed, while being much more accurate (as discussed in Section 6.1, Section 6.2).

## 7 ANALYSIS

To better understand the performance of our KV cache prediction method, and to motivate future optimizations to improve performance, we analyze the quality of the KV cache predictions.

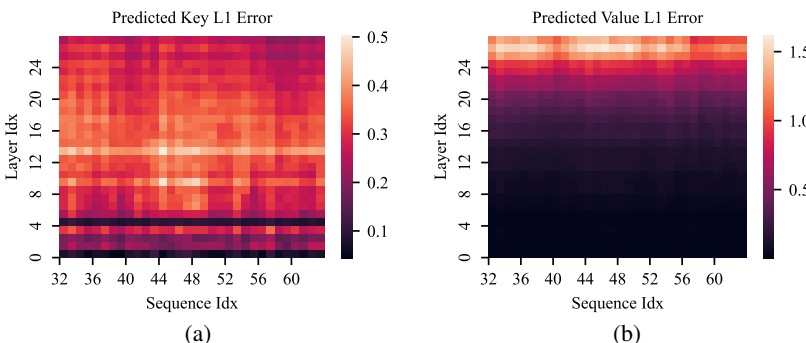

Figure 6: L1 error of predicted keys (Fig. 6a) and values (Fig. 6b) across the layer and sequence dimensions of OpenELM-1.1B-KVP-C-450M.

| Model | arc-c | arc-e | boolq | hellaswag | piqa | sciq | winogrande | Avg | TTFT Reduction |
|---|---|---|---|---|---|---|---|---|---|
| OE 1.1B | 32.34 | 55.43 | 63.58 | 64.81 | 75.57 | 90.60 | 61.72 | 63.43 | 1.0 |
| OE1.1B-KVP-LP-0.75 | **30.97** | **54.42** | **59.79** | **62.91** | **74.92** | **90.40** | **62.04** | **62.21** | 1.34 |
| OE 1.1B-0.75 | 29.52 | 51.81 | 57.52 | 58.38 | 73.88 | 87.70 | 60.46 | 59.90 | 1.34 |
| OE1.1B-KVP-LP-0.50 | **30.80** | **53.75** | 58.17 | **60.35** | **74.32** | **88.90** | **58.96** | **60.75** | 1.93 |
| OE 1.1B-0.50 | 26.28 | 48.15 | **59.79** | 53.74 | 71.33 | 86.10 | 57.46 | 57.55 | 1.93 |
| OE1.1B-KVP-C-450M | **29.01** | **52.53** | **57.95** | **59.20** | **73.01** | **88.00** | **59.43** | **59.88** | 2.36 |
| OE 450M | 27.56 | 48.06 | 55.78 | 53.97 | 72.31 | 87.20 | 58.01 | 57.56 | 2.36 |
| OE1.1B-KVP-LP-0.25 | **27.65** | **46.68** | 56.88 | **54.34** | **72.09** | **85.00** | **55.33** | **56.85** | 3.59 |
| OE 1.1B-0.25 | 24.15 | 41.84 | **60.49** | 43.08 | 68.61 | 82.60 | 52.41 | 53.31 | 3.59 |
| OE1.1B-KVP-C-270M | **28.41** | **48.70** | 53.67 | **55.35** | **71.98** | **87.30** | **57.38** | **57.54** | 3.98 |
| OE 270M | 26.45 | 45.08 | **53.98** | 46.71 | 69.75 | 84.70 | 53.91 | 54.37 | 3.98 |

Table 3: Comparison of OpenELM KV Prediction models with using only the auxiliary model or only the base model for Multiple-Choice Question Answering. We de-emphasize "TTFT Reduction" since the concept doesn't apply to multiple-choice question-answering evaluations.

**L1 Error Across Sequences and Layers:** We analyze the distribution of the L1 loss across layers and sequence indexes in Fig. 6. The errors are relatively stable across sequence index, with a few outliers. Across layers, the magnitude of key error is stable due to OpenELM's usage of key norm (Henry et al., 2020). Value error generally increases with depth due to propagation of errors.

**Multiple-Choice Question Answering:** We analyze the quality of KV cache predictions by running multiple-choice question answering (MCQA) evaluations using the predicted cache as the keys and values for the base model. Since these MCQA evaluations don't produce output tokens, there is no notion of TTFT, and our method doesn't provide a speedup. The purpose of these evaluations is to measure the consistency of the predicted KV cache with the base KV cache through accuracy retention on MCQA.

In Table 3, we present results for 7 MCQA tasks on OpenELM 1.1B and KV Prediction models. We directly compare each KV Prediction model to the results obtained by using only the auxiliary model. In all cases, the KV Prediction model obtains higher average accuracy. We also present MCQA evaluations for OpenELM 3B in Appendix C.

**Density of Differences:** In Appendix D, we analyze the density of KV cache prediction errors. We find that the density of key errors is relatively consistent due to the fact that OpenELM uses key norm. However, the value errors increase in magnitude with network depth.

# 8 CONCLUSION

We present a method for improving time to first token (TTFT) called KV Prediction. Our method uses a small auxiliary model to efficiently predict the KV cache needed by a larger base model. We analyze the theoretical and actual speedup of our model, as well as the accuracy retention. We find that our model maintains a strong efficiency-accuracy trade-off, creating a pareto-optimal trade-off in terms of accuracy retention and TTFT.

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

| Model | Layers Retained From OpenELM1.1B |
|---|---|
| OpenELM1.1B-0.75 | 0, 1, 2, 4, 5, 6, 8, 9, 10, 12, 13, 14, 16, 17, 18, 20, 21, 22, 24, 25, 26 |
| OpenELM1.1B-0.50 | 0, 2, 4, 6, 8, 10, 12, 14, 16, 18, 20, 22, 24, 26 |
| OpenELM1.1B-0.25 | 0, 4, 8, 12, 16, 20, 24 |

(a) Specification of layer-pruned OpenELM 1.1B architectures.

| Model | Layers Retained From OpenELM3B |
|---|---|
| OpenELM3B-0.75 | 0, 1, 2, 4, 5, 6, 8, 9, 10, 12, 13, 14, 16, 17, 18, 20, 21, 22, 24, 25, 26, 28, 29, 30, 32, 33, 34 |
| OpenELM3B-0.50 | 0, 2, 4, 6, 8, 10, 12, 14, 16, 18, 20, 22, 24, 26, 28, 30, 32, 34 |
| OpenELM3B-0.25 | 0, 4, 8, 12, 16, 20, 24, 28, 32 |

(b) Specification of layer-pruned OpenELM 3B architectures.

Table 4: Specification of layer-pruned OpenELM architectures.

## A  LAYER-PRUNED ARCHITECTURES

We provide details of our layer-pruned architectures here. Each architecture is defined by pruning layers at regular intervals from an existing OpenELM architecture. We specify the retained layers in Table 4.

## B  ACCURACY VALUES

We give values used to produce plots. In Table 5, we give accuracies and timing results for OpenELM 1.1B on TriviaQA. In Table 6, we give accuracies for OpenELM 3B on TriviaQA. In Table 7, we give accuracies for OpenELM 1.1B on HumanEval.

## C  OPENELM 3B MCQA EVALUATION

In Section 7, we presented multiple-choice question answering (MCQA) evaluations on KV Prediction models using an OpenELM 1.1B base. We present additional results on OpenELM 3B in Table 8.

## D  DENSITY OF DIFFERENCES IN KV PREDICTION

In Fig. 7, we analyze the distribution of the differences between the predicted KV cache and the target KV cache (e.g. similar to the $\mathcal{L}_C$ introduced in Section 4.3, but without the absolute value being computed). We pass a batch of data through our KV prediction model (to produce predictions $\hat{\mathcal{KV}}_\mathcal{B}$) and through the base model (to produce targets $\mathcal{KV}_\mathcal{B}$), and compute the delta between the predictions and targets at every layer.

We observe that the delta for keys is stable, ranging roughly from -1 to 1 and centered at 0 at all layers. This is due to the fact that OpenELM uses normalized keys, so the distribution of the deltas is relatively consistent. By contrast, the error in predicted values increases with network depth. The distribution remains centered at 0, indicating that our cache prediction method is unbiased.

| Model | FLOPs Reduction (Rel) ↑ | TTFT (s) ↓ | TTFT Reduction (Rel) ↑ | TQA ↑ | TQA (Rel) ↑ |
|---|---|---|---|---|---|
| OE1.1B | 1.00 | 5.59 | 1.00 | 23.57 | 1.00 |
| OE450M | 2.36 | 2.58 | 2.17 | 10.97 | 0.41 |
| OE270M | 3.98 | 1.67 | 3.34 | 7.04 | 0.27 |
| OE1.1B-LP-0.75 | 1.34 | 4.08 | 1.37 | 15.04 | 0.57 |
| OE1.1B-LP-0.50 | 1.93 | 2.78 | 2.01 | 10.13 | 0.38 |
| OE1.1B-LP-0.25 | 3.60 | 1.46 | 3.82 | 4.21 | 0.16 |
| OE1.1B-KVP-C-450M | 2.36 | 3.17 | 1.76 | 16.09 | 0.61 |
| OE1.1B-KVP-C-270M | 3.98 | 2.24 | 2.50 | 11.71 | 0.44 |
| OE1.1B-KVP-LP-0.75 | 1.34 | 4.63 | 1.21 | 20.73 | 0.78 |
| OE1.1B-KVP-LP-0.50 | 1.93 | 3.33 | 1.68 | 17.59 | 0.66 |
| OE1.1B-KVP-LP-0.25 | 3.60 | 2.05 | 2.73 | 10.38 | 0.39 |
| RP-0.75 | 1.33 | 4.31 | 1.30 | 2.61 | 0.10 |
| RP-0.50 | 2.00 | 2.94 | 1.90 | 0.08 | 0.00 |
| RP-0.25 | 4.00 | 4.31 | 1.30 | 0.01 | 0.00 |
| SP-0.50 | 1.33 | 3.73 | 1.50 | 4.29 | 0.16 |
| SP-0.25 | 2.00 | 2.51 | 2.22 | 0.38 | 0.01 |
| L0.75-0.75 | 1.21 | 4.47 | 1.25 | 19.33 | 0.73 |
| L0.75-0.50 | 1.38 | 4.09 | 1.37 | 18.78 | 0.71 |
| L0.75-0.25 | 1.60 | 3.69 | 1.51 | 17.89 | 0.67 |
| L0.75-0.10 | 1.89 | 3.18 | 1.76 | 14.63 | 0.55 |
| L0.75-0.05 | 1.83 | 3.24 | 1.72 | 15.70 | 0.59 |
| L0.75-0.01 | 1.77 | 3.36 | 1.66 | 16.25 | 0.61 |
| L0.50-0.50 | 1.54 | 3.38 | 1.65 | 8.69 | 0.33 |
| L0.50-0.25 | 1.82 | 3.19 | 1.75 | 7.36 | 0.28 |
| L0.50-0.10 | 2.04 | 2.79 | 2.00 | 6.21 | 0.23 |
| L0.50-0.05 | 2.13 | 2.68 | 2.09 | 5.56 | 0.21 |
| L0.50-0.01 | 2.20 | 2.65 | 2.11 | 5.06 | 0.19 |

Table 5: Relative FLOPs, runtime and accuracy values for OpenELM 1.1B on TriviaQA. Values are used to produce Fig. 3 (Left) and Fig. 5a.

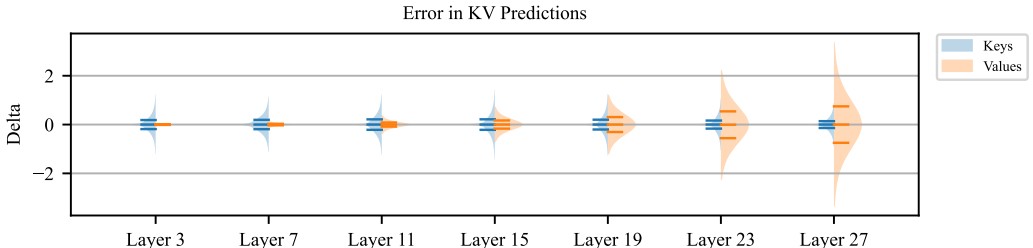

Figure 7: Distribution of the delta between KV cache predictions and targets in 7 different layers of OpenELM-1.1B-KVP-C-450M. The distributions of deltas for keys (blue) and values (orange) are shown separately.

| Model | FLOPs Reduction (Rel) ↑ | TQA ↑ | TQA (Rel) ↑ |
|---|---|---|---|
| OE3B | 1.00 | 40.87 | 1.00 |
| OE1.1B | 2.81 | 23.57 | 0.58 |
| OE450M | 6.64 | 10.97 | 0.27 |
| OE270M | 11.18 | 7.04 | 0.17 |
| OE3B-LP-0.75 | 1.34 | 33.31 | 0.81 |
| OE3B-LP-0.50 | 1.97 | 23.94 | 0.59 |
| OE3B-LP-0.25 | 3.84 | 9.43 | 0.23 |
| OE3B-KVP-C-1.1B | 2.81 | 28.83 | 0.71 |
| OE3B-KVP-C-450M | 6.64 | 15.64 | 0.38 |
| OE3B-KVP-C-270M | 11.18 | 16.70 | 0.41 |
| OE3B-KVP-LP-0.75 | 1.34 | 35.43 | 0.87 |
| OE3B-KVP-LP-0.50 | 1.97 | 30.41 | 0.74 |
| OE3B-KVP-LP-0.25 | 3.84 | 13.11 | 0.32 |
| RP 0.75 | 1.33 | 5.09 | 0.12 |
| RP 0.50 | 2.00 | 0.22 | 0.01 |
| RP0.25 | 4.00 | 0.02 | 0.00 |
| SP 0.50 | 1.33 | 9.45 | 0.23 |
| SP 0.25 | 2.00 | 1.04 | 0.03 |
| L0.75-0.75 | 1.21 | 19.90 | 0.49 |
| L0.75-0.50 | 1.38 | 19.93 | 0.49 |
| L0.75-0.25 | 1.60 | 19.68 | 0.48 |
| L0.75-0.10 | 1.89 | 18.35 | 0.45 |
| L0.75-0.05 | 1.83 | 18.58 | 0.45 |
| L0.75-0.01 | 1.77 | 18.95 | 0.46 |
| L0.50-0.50 | 1.54 | 4.04 | 0.10 |
| L0.50-0.25 | 1.82 | 3.95 | 0.10 |
| L0.50-0.10 | 2.04 | 3.81 | 0.09 |
| L0.50-0.05 | 2.13 | 3.72 | 0.09 |
| L0.50-0.01 | 2.20 | 3.66 | 0.09 |

Table 6: Relative FLOPs and accuracy values of OpenELM 3B on TriviaQA. Values are used to produce Fig. 3 (Right).

| Model | FLOPs Reduction (Rel) ↑ | Pass@1 ↑ | Pass@1 (Rel) ↑ | Pass@10 ↑ | Pass@10 (Rel) ↑ |
|---|---|---|---|---|---|
| OE1.1B | 1.00 | 15.79 | 1.00 | 23.07 | 1.00 |
| OE450M | 2.36 | 8.85 | 0.56 | 14.33 | 0.62 |
| OE270M | 3.98 | 7.41 | 0.47 | 11.51 | 0.50 |
| OE1.1B-LP-0.75 | 1.34 | 1.30 | 0.72 | 17.85 | 0.77 |
| OE1.1B-LP-0.50 | 1.93 | 10.46 | 0.66 | 14.36 | 0.62 |
| OE1.1B-LP-0.25 | 3.60 | 3.37 | 0.21 | 6.94 | 0.30 |
| OE1.1B-KVP-C-450M | 2.36 | 10.12 | 0.64 | 14.37 | 0.62 |
| OE1.1B-KVP-C-270M | 3.98 | 7.20 | 0.46 | 11.68 | 0.51 |
| OE1.1B-KVP-LP-0.75 | 1.34 | 13.82 | 0.88 | 19.83 | 0.86 |
| OE1.1B-KVP-LP-0.50 | 1.93 | 11.51 | 0.73 | 19.03 | 0.82 |
| OE1.1B-KVP-LP-0.25 | 3.60 | 6.24 | 0.39 | 10.73 | 0.47 |
| RP-0.75 | 1.33 | 1.90 | 0.12 | 6.71 | 0.29 |
| RP-0.50 | 2.00 | 0.08 | 0.01 | 0.61 | 0.03 |
| RP-0.25 | 4.00 | 0.00 | 0.00 | 0.00 | 0.00 |
| SP-0.50 | 1.33 | 0.01 | 0.00 | 0.06 | 0.00 |
| SP-0.25 | 2.00 | 0.00 | 0.00 | 0.00 | 0.00 |
| L0.75-0.75 | 1.21 | 9.81 | 0.62 | 15.43 | 0.67 |
| L0.75-0.50 | 1.38 | 10.46 | 0.66 | 15.85 | 0.69 |
| L0.75-0.25 | 1.60 | 10.97 | 0.69 | 16.08 | 0.70 |
| L0.75-0.10 | 1.77 | 10.63 | 0.67 | 15.00 | 0.65 |
| L0.75-0.05 | 1.83 | 10.56 | 0.67 | 14.62 | 0.63 |
| L0.75-0.01 | 1.89 | 6.44 | 0.41 | 12.46 | 0.54 |
| L0.50-0.50 | 1.54 | 2.63 | 0.17 | 5.71 | 0.25 |
| L0.50-0.25 | 1.82 | 2.50 | 0.16 | 5.37 | 0.23 |
| L0.50-0.10 | 2.04 | 1.97 | 0.12 | 4.91 | 0.21 |
| L0.50-0.05 | 2.13 | 2.19 | 0.14 | 5.77 | 0.25 |
| L0.50-0.01 | 2.20 | 0.77 | 0.05 | 3.42 | 0.15 |

Table 7: Relative FLOPs and accuracy values of OpenELM 1.1B on HumanEval. Values are used to produce Fig. 4.

| Model | arc-c | arc-e | boolq | hellaswag | piqa | sciq | winogrande | Avg | TTFT Reduction |
|---|---|---|---|---|---|---|---|---|---|
| OE 3B | 35.58 | 59.89 | 67.40 | 72.44 | 78.24 | 92.70 | 65.51 | 67.39 | 1.0 |
| OE3B-KVP-LP-0.75 | **34.04** | 57.28 | **66.73** | **69.94** | **77.58** | **92.00** | **63.30** | **65.84** | 1.34 |
| OE 3B-0.75 | 33.53 | **59.89** | 65.14 | 67.60 | 76.71 | **92.00** | 62.75 | 65.37 | 1.34 |
| OE3B-KVP-L-0.50 | **33.87** | **56.65** | **64.50** | **67.06** | **76.77** | **90.00** | 59.67 | **64.07** | 1.97 |
| OE 3B-0.50 | 31.06 | 54.67 | 63.36 | 62.43 | 75.95 | 89.50 | **60.54** | 62.50 | 1.97 |
| OE3B-KVP-C-OE1.1B | **33.96** | **57.66** | **64.50** | **66.66** | **76.71** | 90.30 | 61.64 | **64.49** | 2.81 |
| OE1.1B | 32.34 | 55.43 | 63.58 | 64.81 | 75.57 | **90.60** | **61.72** | 63.43 | 2.81 |
| OE3B-KVP-LP-0.25 | **29.27** | **51.39** | **62.63** | **58.87** | **74.59** | 85.30 | 56.27 | **59.76** | 3.84 |
| OE3B-0.25 | 26.88 | 47.10 | 59.69 | 49.89 | 71.33 | **86.00** | **58.56** | 57.06 | 3.84 |
| OE3B-KVP-LP-450M | **30.97** | **54.00** | **62.84** | **61.94** | **74.76** | **88.50** | 57.46 | **61.50** | 6.64 |
| OE450M | 27.56 | 48.06 | 55.78 | 53.97 | 72.31 | 87.20 | **58.01** | 57.56 | 6.64 |
| OE3B-KVP-C-270M | **29.27** | **49.87** | **59.48** | **59.09** | **73.50** | **87.50** | **56.35** | **59.30** | 11.18 |
| OE 270M | 26.45 | 45.08 | 53.98 | 46.71 | 69.75 | 84.70 | 53.91 | 54.37 | 11.18 |

Table 8: Comparison of OpenELM 3B model variants on Multiple-Choice Question Answering. We de-emphasize "TTFT Reduction" since the concept of TTFT doesn't apply to multiple-choice question-answering evaluations.

