# OpenReview forum: "KV Prediction for Improved Time to First Token"
_ICLR.cc/2025/Conference — ICLR 2025 Conference Withdrawn Submission_

### Official Review · Reviewer_S8o8 · 2024-10-30

**Soundness:** 2
**Presentation:** 3
**Contribution:** 2
**Rating:** 5
**Confidence:** 4

**Summary:**

Long TTFT degrades users' experience of edge devices. Therefore, the authors propose KV Prediction, a technique that utilizes an auxiliary model to predict a larger base model's initial value of KV-Cache. The method is evaluated on benchmarks, TriviaQA and HumanEval. On TriviaQA, relative accuracy improvements are in the range of 15%−50% across a range of TTFT FLOPs budgets. On HumanEval Python code completion, accuracy improvements can be up to 30% at fixed TTFT FLOPs budgets. The authors also explore the efficiency-accuracy trade-off space. Real-world validation on an Apple M2 Pro CPU confirms that improvement in FLOPs translates to a TTFT speedup on hardware.

**Strengths:**

1. The algorithm is clearly defined and well explained.
2. The efficiency-accuracy trade-off is well studied through empirical experiments.

**Weaknesses:**

1. At line 066, the paper claims that "We release our code". However, there is no link to the code. Could the authors provide the like or clarify the code release plan?
2. Possible incorrect estimation of FLOPs: In Secture 4.4 Line 303 claims that "The FLOPs-per-token compute cost of transformers inference can be estimated as 2P, where P is the number of parameters in the model". However the caculation in Kaplan et al., 2020 is $C = 2P+2n_{layer}d_{attn}N$ because $n_{ctx}=N$. The second term cannot be omitted. Therefore, the claim that "The total FLOPs required for prompt processing for N tokens is NP" is wrong, and thus Figures 3, 4, and 5 are questionable. Could you address this discrepancy and update the results if necessary?

**Questions:**

1. What is the measured extra memory that Auxilary Model uses?
2. Have the authors tried to co-optimize the algorithm with the Neural Engine on Apple M2 Pro CPU? If not, why? If so, what is the conclusion?

---

> ### Author Response · Authors · 2024-11-19
> **Response to Reviewer S8o8**
>
> Thank you for your feedback. We appreciate your recognition that "The algorithm is clearly defined and well explained", and "**the efficiency-accuracy trade-off is well studied through empirical experiments**".  We address your points below  and hope we can persuade you to increase your rating. Note that line numbers refer to the latest draft.
>
> > 1. At line 066, the paper claims that "We release our code". However, there is no link to the code. Could the authors provide the like or clarify the code release plan?
>
> Our code has already been released publicly. Unfortunately, our code contains copyright notices that identify the authors, which we are not allowed to remove. Thus, we are unable to upload an anonymous version here. We can ask the AC to condition acceptance upon verification of the code release.
>
> > 2. Possible incorrect estimation of FLOPs
>
> Great question. As you point out, the FLOPs estimate in Kaplan et al includes extra terms. However, they also note on page 7 (beneath Table 1, in https://arxiv.org/pdf/2001.08361):
>
> "For contexts and models with dmodel > nctx/12, the context-dependent computational cost per token is a relatively small fraction of the total compute. Since we primarily study models where dmodel >> nctx/12, we do not include context-dependent terms in our training compute estimate."
>
> In our case, taking OpenELM3B as an example, d_model=3072 (on average; it varies by layer), and n_context/12=2048/12=176, an order of magnitude smaller. (And, the evaluations in Fig 3, 4, 5 don't use the full context length, so this is an overestimate of n_context/12).
>
> QUESTIONS ASKED BY THE REVIEWER:
> > Q1: What is the measured extra memory that Auxilary Model uses?
>
> After predicting the KV cache, the auxiliary model is no longer needed. Thus, it never needs to be resident in memory at the same time as the base model. In this scenario, the peak memory is the same as when the base model is used for standard inference. (After the inference, we can load the auxiliary model back in memory to avoid cold starts when the next query comes in.)
>
> In terms of activation tensors, there is no extra overhead. Once the auxiliary model is used, only the predicted KV cache is retained.
>
> We have added a note about memory usage on lines 317-321.
>
> > Q2: Have the authors tried to co-optimize the algorithm with the Neural Engine on Apple M2 Pro CPU? If not, why? If so, what is the conclusion?
>
> We deliberately avoid this because our goal is to assess model performance on edge devices without an accelerator. In other words, the laptop serves as a convenient proxy for studying inference times on lower-end devices that don't have accelerators. We also do not want to unfairly advantage our method by only optimizing our method on the hardware target, as other methods have not been optimized for the hardware target.

---

> ### Author Response · Authors · 2024-11-22
> **Did Our Response Address Your Concerns?**
>
> Hi, we just wanted to check if our comments addressed your concerns, or if you have any followup comments. We hope our response can persuade you to increase your review score, and we look forward to more discussion if you have more comments.

---

> ### Author Response · Authors · 2024-12-01
> **Did Our Response Address Your Concerns?**
>
> Thank you again for the paper review. As the review period is nearly over, we wanted to reach out one more time to make sure we have addressed your comments. We hope you can be persuaded to increase your score.

---

> > ### Comment · Reviewer_S8o8 · 2024-12-01
> >
> > Thank the authors for the response. My concerns have been addressed and I will keep my score.

---

### Official Review · Reviewer_UoQn · 2024-11-02

**Soundness:** 3
**Presentation:** 3
**Contribution:** 2
**Rating:** 5
**Confidence:** 3

**Summary:**

This paper addresses the latency issues in transformer-based language models during inference, particularly the time to first token (TTFT), which is the delay before producing the first output token. In large models, this delay is caused by the prompt processing step, where the model generates the initial output token and creates a key-value (KV) cache for subsequent tokens. This step can be especially slow on edge devices with billion-parameter models, leading to a degraded user experience.
To accelerate TTFT, the authors introduce KV Prediction, a technique that uses a small auxiliary model to precompute an approximation of the KV cache for the base model. This approximated cache enables faster autoregressive generation without repeatedly querying the auxiliary model. The method achieves a pareto-optimal balance of efficiency and accuracy compared to other approaches. For example, in tests on TriviaQA, KV Prediction yields accuracy gains of 15%-50% across various TTFT FLOPs budgets, and on HumanEval for Python code completion, it improves accuracy by up to 30% at fixed FLOPs budgets. Benchmarks on an Apple M2 Pro CPU show that these FLOP savings translate into faster TTFT on real hardware. The authors also plan to release their code to support reproducibility.

**Strengths:**

- Proposes a new method for accelerating the prefilling stage of LLM, using a smaller model
- Gives detailed explanation of the training process.

**Weaknesses:**

- The method is only tested on OpenELM family models, thus the generalizability remains a question. This is the major concern.
- For the CPU inference, shouldn’t we also consider TPOT? From Table 3, there are some improvements over auxiliary-only and base models, but they also improve TPOT. With this consideration, is low TTFT but high TPOT worthy? How about the total execution time?
- For HumanEval, the advantages are not as obvious as in TriviaQA, as shown in Figure 4. This makes the generalizability of the proposed method on a broad range of benchmarks remains a question. Why not test on more datasets?

**Questions:**

The problems raised in weakness

---

> ### Author Response · Authors · 2024-11-19
> **Response to Reviewer UoQn**
>
> Thank you for the review. We appreciate your recognition that we propose "a new method for accelerating the prefilling stage of LLM", and that we "give a detailed explanation of the training process". We address your points below  and hope we can persuade you to increase your rating. Note that line numbers refer to the latest draft:
>
> > 1. The method is only tested on OpenELM family models, thus the generalizability remains a question. This is the major concern.
>
> We focused on giving a few different evaluations (generative QA, multiple-choice question answering) and fine-tuning for code completion (HumanEval) to test for generality. As most Transformer architectures are similar (with relatively minor differences such as prenorm/postnorm, etc), we are hoping to persuade you of Review kKHv's perspective ("This paper conducts extensive experiments") and S8o8's perspective ("The efficiency-accuracy trade-off is well studied through empirical experiments.").
>
> > 2. For the CPU inference, shouldn’t we also consider TPOT?
>
> As shown in Figure 1 (right), TTFT can climb to 10s of seconds in compute-bound scenarios. Users are not willing to wait this long for outputs (as discussed on lines 132-137). As shown in Figure 1 (left), TTFT can take orders of magnitude longer than autoregressive outputs. We focus on TTFT because it dictates the responsiveness of an application, and since users can begin consuming autoregressive outputs after the first token is emitted.
>
> > 3. For HumanEval, the advantages are not as obvious as in TriviaQA, as shown in Figure 4.
>
> We believe Figure 4 (HumanEval) to demonstrate strong results. For example, our method improves accuracy by roughly 10% (relative) at 2x FLOPs reduction for Pass@1, and by roughly 30% (relative) at 2x FLOPs reduction for Pass@1. Although the TriviaQA results are even more impressive, we believe HumanEval to represent a significant boost in accuracy as well. We are hoping the results on TriviaQa (Fig 3), HumanEval (Fig 4), and multiple-choice question answering (Table 3, Table 8) persuade you of the efficacy of our method.

---

> ### Author Response · Authors · 2024-11-22
> **Did Our Response Address Your Concerns?**
>
> Hi, we just wanted to check if our comments addressed your concerns, or if you have any followup comments. We hope our response can persuade you to increase your review score, and we look forward to more discussion if you have more comments.

---

> ### Author Response · Authors · 2024-12-01
> **Did Our Response Address Your Concerns?**
>
> Thank you again for the paper review. As the review period is nearly over, we wanted to reach out one more time to make sure we have addressed your comments. We hope you can be persuaded to increase your score.

---

### Official Review · Reviewer_kKHv · 2024-11-02

**Soundness:** 2
**Presentation:** 3
**Contribution:** 2
**Rating:** 5
**Confidence:** 3

**Summary:**

This paper introduces a novel method to reduce the time to first token.  It uses a small auxiliary model to predict the KV cache for the base model. Experiments on TriviaQA and humaneval demonstrates the effectiveness in achieving a pareto-optimal efficiency-accuracy trade-off when compared to existing baselines.

**Strengths:**

Reducing the time to the first token is a crucial and challenging problem. This paper conducts extensive experiments and makes significant efforts to design KV prediction, which demonstrates effectiveness in achieving a Pareto-optimal trade-off between efficiency and accuracy.

**Weaknesses:**

1. Based on the experimental results, the base model performance can be easily affected after applying the proposed method.  Compared to other methods that can indirectly reduce time to the first token, such as quantization, is the Pareto curve of KV prediction still considered optimal?

2. Could the authors provide additional performance results on challenging reasoning tasks (e.g., GSM8K/MATH) to assess whether this method impacts the base model's performance?

**Questions:**

see the weaknesses section.

---

> ### Author Response · Authors · 2024-11-19
> **Response to Reviewer kKHv**
>
> Thank you for the review. **We appreciate your highlighting of our extensive experiments and our strong efficiency-accuracy trade-off.** We address your points below and hope we can persuade you to increase your rating. Note that line numbers refer to the latest draft.
>
> > 1. Compared to other methods that can indirectly reduce time to the first token, such as quantization, is the Pareto curve of KV prediction still considered optimal?
>
> Great question. We use "shrinking the model" (rather than quantization) as a strong baseline to test this. As you point out, if you switch from an OpenELM1.1B model to an OpenELM 450M model, you expect to see an improvement in TTFT and a reduction in accuracy. The question you're asking is, "how does shrinking the model compare to using our method with a smaller auxiliary model"?
>
> This comparison is visible in Figure 3 and 4. For example, in Figure 3 (left), our model with an Auxiliary of size 450M (green) obtains ~50% better accuracy (relative) compared to a baseline OpenELM 450M model (blue), at the same TTFT. Our accuracy loss at extreme FLOPs reductions (green) outperforms the efficiency-accuracy trade-off of simple model scaling (blue). We outperform the strong baseline of "shrinking the model."
>
> To address your point about quantization: We do not compare with quantization, since quantization is a generically applicable method that could be applied to every model (including ours). Since it would speed up all models, we avoid adding this experimental complexity.
>
> > 2. Could the authors provide additional performance results on challenging reasoning tasks (e.g., GSM8K/MATH)
>
> We appreciate your suggestion. We attempted evaluations for these datasets, but we found that OpenELM baselines performed very poorly (roughly 1% accuracy). Note that these models are very small (~1B or ~3B), and generally small models do not perform well at reasoning. Additionally, our OpenELM baselines have only gone through pretraining (no SFT or RLHF). We hope you find the existing experiments (TriviaQA, code completion, multiple-choice question answering) sufficient, and hope you agree with Reviewer S8o8 ("The efficiency-accuracy trade-off is well studied through empirical experiments").

---

> ### Author Response · Authors · 2024-11-22
> **Did Our Response Address Your Concerns?**
>
> Hi, we just wanted to check if our comments addressed your concerns, or if you have any followup comments. We hope our response can persuade you to increase your review score, and we look forward to more discussion if you have more comments.

---

> ### Author Response · Authors · 2024-12-01
> **Did Our Response Address Your Concerns?**
>
> Thank you again for the paper review. As the review period is nearly over, we wanted to reach out one more time to make sure we have addressed your comments. We hope you can be persuaded to increase your score.

---

> > ### Comment · Reviewer_kKHv · 2024-12-01
> > **Thank you for your rebuttal**
> >
> > Thank you for your rebuttal. I have also reviewed the discussions with other reviewers and have decided to maintain my current rating.

---

### Official Review · Reviewer_QA3R · 2024-11-12

**Soundness:** 1
**Presentation:** 1
**Contribution:** 2
**Rating:** 3
**Confidence:** 4

**Summary:**

The paper uses a smaller model based approach to predict the KV cache content to reduce the TTFT and the improve the prefill stage latency of LLM serving.

**Strengths:**

The idea of KV cache prediction based on Auxiliary model seems unique, though there are potential issues (as described in the weakness).

**Weaknesses:**

1. The code has not been released yet! However, the authors claimed that as a part of their contributions. Anyway, releasing code can not be inferred as a technical contribution.

2. The paper mainly focuses on works of token eviction to justify the claim of TTFT increase. However, there are works like KV cache quantization example, KIVI [1], GEAR [2], for which this may not be true always under all settings. Additionally, for complex reasoning tasks KV quantization have shown more promise as opposed to eviction [2], so please do compare with such approaches!

3. There are system level works like Sarathi [3], and Etalon [4] that tried to reduce the prefill and decode times through lossless approaches. The authors referred to orthogonal approaches like pruning, but not these relevant lossless approaches!

4. There are existing works that tried to reduce TTFT, including Minference [5], and SeerAttention [6]. Please compare your work with them.

5. The method would increase the memory data transfer for the KV cache as opposed to the baseline.

6. The method would increase the memory requirement due to storage need of an auxiliary model.

7. The results are not comprehensive, please demonstrate on popular LLMs like LLaMA, OPT, Phi etc. on tasks like PG19, GSM8k-5shot, XSUM, passkey retrieval to understand the merit of the proposed approach.

8. The approach requires training to do KV compression which might not be feasible on device, and most of the existing KV compression settings are assumed to be training free due to the underlying assumption of it to be an inference time optimization. Brining training in the loop would significantly limit its application as such assumptions wont be valid.

[1] KIVI: A Tuning-Free Asymmetric 2bit Quantization for KV Cache, ICML 2024.

[2] GEAR: an efficient kv cache compression recipe for near-lossless generative inference of LLM, arxiv 2024.

[3] Sarathi: efficient llm inference by piggybacking decodes with chunked prefills, USENIX 2024.

[4] Etalon: Holistic Performance Evaluation Framework for LLM Inference Systems, arxiv 2024

[5] Minference 1.0: Accelerating pre-filling for long-context LLMs via dynamic sparse attention, NeurIPS 2024.

[6] SeerAttention: Learning Intrinsic Sparse Attention in Your LLMs, arxiv 2024

**Questions:**

Please refer to weakness.

---

> ### Author Response · Authors · 2024-11-19
> **Response to Reviewer QA3R (Part 1)**
>
> Thank you for your review. We appreciate the detailed feedback, as well as the pointers to related works. Your thoroughness is appreciated.
>
>  We believe that much of the criticism stems from a misunderstanding of our contribution. We emphasize that our goal is to improve on-device TTFT, for which the suggested additional baselines do not apply:
> 1. **Our goal is NOT to compress the KV cache**. Instead, our goal is to generate the KV cache more rapidly (in other words, to improve TTFT). Other works that focus on KV cache compression (KIVI, GEAR) are not suitable baselines since they do not improve TTFT.
> 2. **We do NOT focus on server-side inference** - instead, we focus on on-device inference. These scenarios are very different, as server-side works generally assume multiple hosts with lots of memory, or a pipelined input processing scenario. Thus, we do not compare with server-side works such as Minference and SeerAttention (see details below).
> 3. We focus on **on-device inference, NOT long-context inference**. As shown in Fig 1, on-device inference exhibits high TTFT even at relatively modest context lengths, which motivates our study. Thus, we do not compare with long-context methods or evaluate on long-context datasets, since running 32k tokens on device is not feasible (as seen by extrapolating Fig 1).
>
> We address your points below. We hope to have appropriately addressed your feedback towards an improved rating. Note that line numbers refer to the latest draft.
>
> > 1. The code has not been released yet. Releasing code can not be inferred as a technical contribution.
>
> We understand your perspective, and have moved the mention of the code release out of the list of contributions.
>
>  We believe releasing code is an important contribution for reproducibility. Although one can argue that it is not strictly a scientific contribution, it is important for allowing others to understand every detail of a work. In acknowledgment of your point that it isn't a scientific contribution, we have moved our mention of the code release out of the list of contributions (it's in the same pargraph, but is not numerically listed as a scientific contribution).
>
> Our code has already been released publicly. Unfortunately, our code contains copyright notices that identify the authors, which we are not allowed to remove. Thus, we are unable to upload an anonymous version here. We can ask the AC to condition acceptance upon verification of the code release.
>
> > 2. The paper mainly focuses on works of token eviction to justify the claim of TTFT increase.
> We believe you are referring to lines 100-102, in which we mention that prompt compression methods increase TTFT.
>
> We agree that KIVI and GEAR would not increase TTFT. However, *they also do not improve TTFT* (they leave it unchanged), as they are focused on making the KV cache more compact, rather than generating it more quickly. *This is why we do not provide empirical comparisons* with KIVI and GEAR compared to our method.
>
> We have added KIVI and GEAR to related works (line 96-99), thank you for the recommendation.
>
> > 3. There are system level works like Sarathi [3], and Etalon [4] that tried to reduce the prefill and decode times through lossless approaches.
>
> We discuss system-level approaches for reducing prefill times in lines 82-90 (Server-Side TTFT Efficiency). We appreciate the suggestion of adding Sarathi and Etalon to this section, and we have added them.
>
> However, **our focus is on on-device TTFT, not server-side TTFT**. We are trying to improve long TTFTs that appear on-device (Fig 1). Server-side techniques are usually not applicable to on-device (e.g. server-side techniques usually require storing KV caches on high-memory nodes, access to efficient custom GPU kernels, etc). In particular, Sarathi studies multi-gpu pipelined inference on servers, and Etalon studies inference performance of various methods when diverse prompt lengths hit a server at different times. Thus, we do not compare with them, since our focus is on-device TTFT.
>
> > 4. Compare to Minference and SeerAttention.
>
> Minference and SeerAttention both focus on long context. **By contrast, our work focuses on on-device inference** on edge devices without an accelerator (our on-device evaluations use CPU). As shown in Figure 1, it is not feasible to run with such long context (~32k) on device.
>
> Additionally, Minference requires optimized GPU kernels to obtain a speedup, whereas our setting assumes an edge device without a powerful accelerator. Likewise, SeerAttention requires an optimized sparse attention kernel. Our focus is on on-device inference, in which a GPU is not available.
>
> We added these two works to our related work section, thank you for the recommendation.

---

> ### Author Response · Authors · 2024-11-19
> **Response to Reviewer QA3R (Part 2)**
>
> > 5. The method would increase the memory data transfer for the KV cache as opposed to the baseline
>
> Our method generates a small auxiliary KV cache, then uses it to predict the base KV cache. Once the predicted KV cache is generated, the model runs in generation mode, in a manner identical to standard inference. We are not clear on what memory transfer you are concerned about. Can you please clarify?
>
>  We provide timing experiments (Section 6.3) that empirically demonstrate that our method is faster than baselines, which we hope mitigates your worries about any overhead from our method. We empirically verify that KV cache prediction is fast.
>
> > 6. The method would increase the memory requirement due to storage need of an auxiliary model.
>
> The number of deployed model parameters is indeed greater. Although, the auxiliary model can be unloaded after prompt processing, then loaded back into memory after the query is processed (so the system is ready for the next query, and a cold start is not required). Thus, the peak memory used by the system will not increase. We added an explicit note of this on lines 317-321 to make sure readers are clear on this.
>
> We believe the most important metric to the end user is the overall runtime, which we improve. Also, note that the auxiliary model can be moved out of memory during the generation phase, as it is not needed.
>
> > 7. The results are not comprehensive
>
> We present results on generative question answering (Fig 3), code completion (Fig 4), and multiple-choice question answering (Table 3, 8). Please note that our focus is on on-device execution, not long-context, which is why we don't evaluate on long-context datasets. Even at standard context lengths, on-device TTFT becomes very high (Fig 1).
>
> We hope you can be persuaded by the perspective of Reviewer kKHv ("This paper conducts extensive experiments") and Reviewer S8o8 ("The efficiency-accuracy trade-off is well studied through empirical experiments").
>
> > 8. The approach requires training to do KV compression which might not be feasible on device, and most of the existing KV compression settings are assumed to be training free due to the underlying assumption of it to be an inference time optimization.
>
> First, we emphasize that **our method is not about KV compression**. Instead, our method is improving TTFT through predicting the KV cache.
>
> Second, we respectfully disagree with the reviewers perspective that TTFT improvement methods (or orthogonal works on KV compression) should be training-free.
> - **Many inference-time optimizations in the literature are not training-free**. For instance, pruning and quantization methods are usually training-aware, or require fine-tuning at the very least (e.g. GPTQ, AWQ, etc.). Inference time optimizations are not generally assumed to be training free.
> - **Prior works such as SeerAttention (again not directly comparable to ours, but you mention it as a related work) do in fact require fine-tuning.**
> - The fact that prior works have studied training-free methods for improving TTFT should not preclude future works from studying training methods for improving TTFT.
>
> Furthermore, we respectfully disagree with the statement that other existing inference-time methods are easier to deploy. Other inference-time methods for improving TTFT often require specialized GPU kernel implementations (e.g. Minference, SeerAttention), which can reduce portability to other hardware platforms. By contrast, our model can be deployed on-device using only standard inference tooling (e.g. any standard software that converts neural networks to on-device formats, e.g. TorchScript, etc.).

---

> ### Author Response · Authors · 2024-11-22
> **Did Our Response Address Your Concerns?**
>
> Hi, we just wanted to check if our comments addressed your concerns, or if you have any followup comments. We hope our response can persuade you to increase your review score, and we look forward to more discussion if you have more comments.

---

> ### Comment · Reviewer_QA3R · 2024-11-26
> **Response to authors**
>
> In L144-145 authors mentioned on increase in prompt length and/or batch size, however, for on device use classes larger batch-size would not be that realistic. And the authors did not intend to focus on larger prompt length as well. So, I am a bit confused here.
>
> The author should have results on longbench, as their example of 1024 prompts length (L146-147), should match that of longbench. Also, gsm8k has prompt length of similar. Demonstrating on LM harness only should not suffice. So the evaluations remain weak. However, in view of some clarifications from the authors, at least clarifying the fact that they ONLY focus on on-device ttft improvement, I raise the score.
>
> The assumption of on device use case mode generally conflicts with the set up where they need a separate auxiliary model, asking for more memory.
>
> That additionally need training of the auxiliary model and predictor. So, I am not sure how this work would motivate plug and play deployment for on device ttft improvement. The training sensitivity,, data scarcity,, data privacy, possibility of training of aux model and predictor keeps me thinking on the real use cases of this approach.
>
> I thus in summary, still am not convinced on the paper's contribution and believe the set up needs to have a care rethinking. Or surely more clarification and eval are needed.

---

> ### Author Response · Authors · 2024-11-28
> **Response to “Response to authors” 1/2**
>
> Thank you for your reply. We appreciate the interactive discussion and you are bringing up good discussion points. We address them below. We hope we can persuade you to further increase your score, and we look forward to more followup discussions.
>
> > In L144-145 authors mentioned on increase in prompt length and/or batch size, however, for on device use classes larger batch-size would not be that realistic. And the authors did not intend to focus on larger prompt length as well. So, I am a bit confused here.
>
> **We focus on context lengths in the realm of hundreds of tokens.** Even at this context length, TTFT is significant (Fig. 1). Please note that **what the community considers to be “long context” is usually 4k+, which is far longer than the context lengths we discuss here.** We do not focus on long context.
>
> Regarding batch size:
> - **Even at Batch Size 1**, on-device TTFT is significant. For example, see Fig 1: at prompt length 512, TTFT is 2.5 seconds. This is evaluated on a modern laptop without an accelerator, but still with a relatively expensive CPU. **The TTFT would be even higher on an inexpensive edge device with a less performant CPU** (we use a Macbook Pro CPU since it is easier to interact with and widely available for comparisons. As noted in the paper, we do not use the on-device AI accelerator).
> - **Several techniques can make use of batch size > 1 on-device.** Example 1: in code completion software, completions are often generated in advance for multiple different locations where the user might want to generate code. This involves processing different prompts in a single batch. Example 2: For question-answering tasks augmented by retrieval, we can process documents in a batched manner (as described in Superposition Prompting: https://arxiv.org/abs/2404.06910).
>
> > The author should have results on longbench, as their example of 1024 prompts length (L146-147), should match that of longbench.
>
> **LongBench uses context lengths far longer than 1024 - most context lengths are above 4096. Our focus is on context lengths of a few hundred tokens.** We reiterate that our goal is not exploring long-context datasets such as LongBench. **A histogram showing the context lengths and their relative frequencies is presented at https://github.com/THUDM/LongBench . Please note the very long context lengths used in LongBench, up to 32k.**
>
> Handling such context lengths requires special methods. This is beyond our scope. We find that **even the pretrained baseline OpenELM model does not perform well on LongBench, as expected, as it was not trained or fine-tuned for long contexts, so we do not present results for baselines or our method.**
>
> > Also, gsm8k has similar prompt length
>
> We appreciate your suggestion. We attempted evaluations for GSM8k and MATH datasets, but we found that OpenELM baselines performed very poorly (roughly 1% accuracy). Note that these models are very small (~1B or ~3B), and generally small models do not perform well at reasoning. Additionally, our OpenELM baselines have only gone through pretraining (no SFT or RLHF). We hope you find the existing experiments (TriviaQA, code completion, multiple-choice question answering) sufficient, and hope you agree with Reviewer S8o8 ("The efficiency-accuracy trade-off is well studied through empirical experiments").
>
> > The assumption of on device use case mode generally conflicts with the set up where they need a separate auxiliary model, asking for more memory.
>
> We discuss memory usage in 316-321. The extra memory used is generally a fraction of that used by the base network. We believe that the metric that the end user cares about most is the time spent waiting for the query to be answered (lines 134-136). If a small amount of extra memory is available, then a technique that can make use of it to reduce TTFT is valuable.

---

> > ### Author Response · Authors · 2024-11-28
> > **Response to “response to authors” 2/2**
> >
> > > That additionally need training of the auxiliary model and predictor. So, I am not sure how this work would motivate plug and play deployment for on device ttft improvement.
> >
> > **Most methods for deploying on-device models requires retraining. Our model is no different.**
> > **Here are examples** of model deployment techniques, **all of which require retraining**:
> > - **Small Model Design:** Training smaller models (e.g. Llama 3.2 (https://ai.meta.com/blog/llama-3-2-connect-2024-vision-edge-mobile-devices/), OpenELM (https://arxiv.org/abs/2404.14619), TinyLlama (https://github.com/jzhang38/TinyLlama), all requires training. **Our model is no different - we have a new network design that makes use of a small auxiliary model and a pretrained base model.** Just like other works on model design, we need to train our model.
> > - **Quantization:** General quantization techniques, as well as those specifically evaluated on LLMs (GPTQ [https://arxiv.org/abs/2210.17323], AWQ (https://arxiv.org/abs/2306.00978), and many more) require using data to quantize the network.
> > - **Pruning:** Pruning techniques (LLM In Flash (https://arxiv.org/abs/2312.11514), QSparse (https://arxiv.org/html/2407.10969v2)) use data.
> >
> > **Making use of training data to develop an efficient model is a standard practice.** It is not unique to our work.
> >
> > > The training sensitivity,, data scarcity,, data privacy, possibility of training of aux model and predictor keeps me thinking on the real use cases of this approach.
> >
> > **Our method only uses the same pretraining data as our OpenELM baseline models. This consists of standard widely-available web text data (RefinedWeb). There is no extra data gathered, thus there are no issues of data scarcity/privacy. Only standard LLM pretraining datasets are used.**

---

### Comment · Area_Chair_HS7H · 2024-11-24

Dear Reviewers,

This is a gentle reminder that the authors have submitted their rebuttal, and the discussion period will conclude on November 26th AoE. To ensure a constructive and meaningful discussion, we kindly ask that you review the rebuttal as soon as possible and verify if your questions and comments have been adequately addressed.

We greatly appreciate your time, effort, and thoughtful contributions to this process.

Best regards,
AC

---

### Note · Authors · 2024-12-17

I have read and agree with the venue's withdrawal policy on behalf of myself and my co-authors.